# Effect of Sulfate Ions on Galvanized Post-Tensioned Steel Corrosion in Alkaline Solutions and the Interaction with Other Ions

**DOI:** 10.3390/ma15113950

**Published:** 2022-06-01

**Authors:** Andrés Bonilla, Cristina Argiz, Amparo Moragues, Jaime C. Gálvez

**Affiliations:** Departamento de Ingeniería Civil, Construcción, E.T.S de Ingenieros de Caminos, Canales y Puertos, Universidad Politécnica de Madrid, c/Profesor Aranguren 3, 28040 Madrid, Spain; af.bonilla@alumnos.upm.es (A.B.); cg.argiz@upm.es (C.A.); amparo.moragues@upm.es (A.M.)

**Keywords:** corrosion current density, sulfate, galvanized steel, alkaline solutions, linear polarization resistance, electrochemical impedance spectroscopy

## Abstract

Zinc protection of galvanized steel is initially dissolved in alkaline solutions. However, a passive layer is formed over time which protects the steel from corrosion. The behavior of galvanized steel exposed to strong alkaline solutions (pH values of 12.7) with a fixed concentration of sulfate ions of 0.04 M is studied here. Electrochemical measurement techniques such as corrosion potential, linear polarization resistance and electrochemical impedance spectroscopy are used. Synergistic effects of sulfate ions are also studied together with other anions such as chloride Cl^−^ or bicarbonate ion HCO_3_^−^ and with other cations such as calcium Ca^2+^, ammonium NH_4_^+^ and magnesium Mg^2+^. The presence of sulfate ions can also depassivate the steel, leading to a corrosion current density of 0.3 µA/cm^2^ at the end of the test. The presence of other ions in the solution increases this effect. The increase in corrosion current density caused by cations and anions corresponds to the following orders (greater to lesser influence): NH_4_^+^ > Ca^2+^ > Mg^2+^ and HCO_3_^−^ > Cl^−^ > SO_4_^2−^.

## 1. Introduction

The phenomenon of corrosion usually causes more severe damage to prestressed steel structures than to conventional reinforced concrete structures. Protection of prestressed galvanized steel wires is ensured by injecting alkaline grout into polyethylene ducts covering the strands. However, in areas not protected by these ducts, in deteriorated places, or in areas with insufficient grout, corrosion phenomena can occur. Accumulation in these areas of water contaminated with aggressive ions coming from the atmosphere, marine environments, with decomposition products of organic matter can cause corrosion of these wires and failures in the post-tensioned strands.

Galvanizing protects steel through two mechanisms. Firstly, it creates a physical barrier that isolates it and acts as a sacrificial anode. In addition, corrosion products create a second protective barrier. The behavior of zinc in alkaline media has already been considered in the literature [1,2,3]. Zinc in contact with the alkaline matrix of cement in its fresh state shows temporary chemical instability. High pH values of the aqueous phase inside concrete pores, usually above 12.5, cause zinc oxidation. The cathodic reaction is associated with water hydrolysis and generates hydrogen on the galvanized surface, according to Equation (1).
(1)$$2H_{2}O+2e^{-}\to 2{\mathrm{OH}}^{-}+H_{2\left(g\right)}$$

The possible transformation of molecular hydrogen into physically adsorbed atomic hydrogen, proposed by Riecke [4], increases the risk of hydrogen embrittlement in galvanized post-tensioned steel.

A Pourbaix diagram of zinc indicates that, at a pH value around 12, it forms an insoluble oxide layer of ZnO/Zn(OH)_2_ more stable than the oxide layer formed at pH 13. With high alkalinities, Zn has an amphoteric behavior, forming soluble ions Zn(OH)_3_^−^ and Zn(OH)_4_^2−^ [5]. Formation of Zn(OH)_2_ leads to hydrogen formation [6]:(2)$$\mathrm{Zn}+2H_{2}O\to \mathrm{Zn}{\left(\mathrm{OH}\right)}_{2}+H_{2\left(g\right)}$$

The risk of corrosion in an alkaline medium and in the presence of calcium ions can be limited due to the formation of a passive layer of calcium hydroxyzincate Ca(Zn(OH)_3_)_2_ · 2H_2_O, which is stable and protective. Some authors [7,8] have identified a value of pH 13.3 ± 0.1 as the limit for the passivation of galvanized steel. At a lower value of pH than 13.3, Ca(Zn(OH)_3_)_2_ · 2H_2_O crystals are small enough to form a thin, homogeneous and stable layer on the surface of the steel capable of keeping it passive. At a higher value of pH than 13.3 and when the calcium content is low, the size of the crystals increases, making it difficult to cover the entire surface of the galvanized steel. In this case, large, isolated Ca(Zn(OH)_3_)_2_ crystals that do not passivate galvanized steel are formed.

Other works [9] have studied the behavior of galvanized steel as a function of pH in the absence of Ca^2+^ ions. In the range of 12 < pH < 12.8, the galvanized layer dissolves at a slow speed. In the range of 12.8 < pH < 13.3, the galvanized layer is capable of being covered with a protective layer that insulates it. However, at pH > 12.8 ± 0.1, hydrogen release occurs. At a value of pH > 13.3, the galvanizing layer dissolves completely. It is worth noting that the role of sulfate ions in the corrosion of galvanized steel has been less studied. Acha [10] studied stress corrosion of prestressed steel immersed in saturated solutions of Ca(OH)_2_ with five concentrations of sulfate ions at various values of pH (0.01 M SO_4_^2^^−^ at a pH of 12.1; 0.025 M SO_4_^2^^−^ at a pH of 12.2; 0.05 M SO_4_^2^^−^ at a pH of 12.4; 0.1 M and 0.2 M SO_4_^2^^−^ at a pH of 12.85). Results showed a limiting sulfate concentration between 0.025 (pH = 12.2) and 0.05 (pH = 12.4). Above this limit, the steel surface presented severe localized corrosion, and below this concentration limit, the steel remained passive. Liu et al. [11] also showed that a sulfate concentration of between 0.02 and 0.03 mol/L, in a saturated solution of Ca(OH)_2_, produced steel corrosion.

Therefore, corrosion of prestressed steel in the presence of sulfates depends on the sulphate ion concentration in the solution and on the pH. Carsana and Bertolini [12] identified a pH dependence on the anodic behavior of steel in sulphate solutions for the corrosion of the steel. Acha’s thesis also addressed the effect of bicarbonate ions (0.05 M concentration combined with pH = 11 and 0.1 M concentration with pH = 8.2) and of carbonate ions (CO_3_^2−^) in saturated solutions of Ca(OH)_2_. In both cases, current density reaches values lower than 0.2 µA/cm^2^ after 45 days. These ions do not cause corrosion problems in prestressed steel.

In alkaline media and in the presence of carbonates, the most common corrosion product is hydrozincite (Zn_5_(CO_3_)_2_(OH)_6_) [13,14], and in the presence of sulfates, zinc hydroxysulfate (Zn_4_(SO_4_)(OH)_6_ · 3H_2_O). After galvanized steel exposure to marine environments, the formation of a passive layer of hexagonal crystals of simonkolleite Zn_5_Cl_2_(OH)_8_ · H_2_O, and zincite ZnO has been identified as the main corrosion product. Simonkolleite is formed after hydrozincite, both being white crystalline compounds. Later, a more protective layer of gordaite NaZn_4_Cl(OH)_6_SO_4_ · 6H_2_O can be formed by incorporation of sulfate and sodium ions in the crystalline structure of simonkolleite. Soluble compounds such as ZnCl_2_ and ZnSO_4_ have also been identified in marine environments [13,15,16,17,18,19].

Xu et al. [20] studied the effect of cations from different sulfate salts (MgSO_4_, (NH_4_)_2_SO_4_, Na_2_SO_4_, CaSO_4_) added in a fixed concentration of 0.01 mol/l. The corrosion study was carried out in saturated calcium hydroxide solutions. Solutions of magnesium sulfate and ammonium sulfate showed higher rates of corrosion than solution with sodium sulfate. Solution pH was lowered with the addition of ammonium sulfate and magnesium sulfate. Neupane et al. [21] studied the effect of NH_4_^+^, Na^+^, and Mg^2+^ cations on the corrosion of galvanized steel. Solutions of (NH_4_)_2_SO_4_, Na_2_SO_4_ and MgSO_4_ 0.5 M were prepared in distilled water. The increase in corrosion current density caused by cations and anions corresponds to the following order (greater to lesser influence): Na^+^ > NH_4_^+^ > Mg^2+^. Magnesium ions form finer, more compact and less porous corrosion products than the other salts.

There are numerous studies on the influence of potentially aggressive ions on the corrosion of galvanized steel. The results obtained show that the critical concentrations for each ion are often determined by the pH value of the solution. However, the effect that the joint presence of two or more potentially aggressive ions generates has not yet been determined.

In the present work, we studied the effect of sulfate ions on galvanized steel in alkaline solution and the synergistic effect of sulfate ions with various cations and anions found in seawater or marine environments (Ca^2+^, NH_4_^+^, HCO_3_^−^, Mg^2+^ and Cl^−^), or in the atmosphere, and their influence on the corrosion of the galvanized steel wires.

## 2. Experimental Work

### 2.1. Solutions

Concentrations of different ions were used, based on those obtained in real solutions produced by the action of rainwater together with degradation processes of living beings’ waste. Table 1 indicates the ionic composition of six synthetic solutions prepared from the following salts: Na_2_SO_4_, Ca(OH)_2_, NH_4_COOH, NaHCO_3_, Mg(COOH)_2_ and NaCl. Solution pH was set at 12.7 by adjusting with NaOH 2M. The main component was the sulfate ion, followed by magnesium, chloride and ammonium. Ions were introduced incrementally in order to determine which ones were responsible for corrosion initiation.

### 2.2. Corrosion Electrochemical Cells

Electrochemical cells were made in polypropylene bottles using a three-electrode system (Figure 1). A Ag/AgCl electrode was used as a reference electrode. Stainless steel mesh was used as a counter electrode and galvanized wire (nominal diameter of 0.519 cm) as a working electrode. Galvanized wires were cleaned with alcohol. Adhesive tape was used for limiting an exposed attack area of 4891 cm^2^. To avoid carbonation, the solution was covered with a liquid paraffin layer.

### 2.3. Techniques

#### 2.3.1. Electrochemical Tests

Electrochemical tests were carried out with an Autolab PGSTAT 204 potentiostat/galvanostat from MetrohmAutolab BV^®^. NOVA 2.4.1 software with FRA 32 impedance module (Figure 1) was used. Linear polarization resistance (LPR) and electrochemical impedance spectroscopy (EIS) were also used. The LPR method was used to determine the instantaneous corrosion rate [22,23]. Measurement was carried out by applying a polarization scan from −20 mV to 20 mV around the open circuit potential (OCP) at a sweep rate of 0.1667 mV/s. Ohmic drop $\left(R_{\Omega }\right)$ obtained by an electrochemical impedance technique is then subtracted from this resistance. Therefore, charge transfer resistance between zinc surface and solution $\left(R_{p}\right)$ (Equation (3)) is calculated as:(3)$$R_{p}=R_{p\;\left(\mathrm{LPR}\right)}-R_{\Omega \;\left(\mathrm{EIS}\right)}$$

Corrosion current density (I_corr_) was obtained from the polarization resistance (R_p_) calculated as the slope of the polarization resistance curve around the corrosion potential according to the Stern and Geary relationship (Equation (4)) [24] with parameter B = 13 [8,25,26,27] and the procedure proposed in UNE 112072 standard [22].
(4)$$I_{\mathrm{corr}}=\;B\cdot \frac{1}{R_{p}\cdot A}$$

Impedance measurements (EIS) were carried out by potentiostatic control in a frequency range between 10 mHz and 100 Khz, taking 10 points per decade. Amplitude of the input AC voltage signal was ±10 Mv (rms). This technique consists of taking measurements by applying a small signal of alternating current and constant voltage to a working electrode, making frequency sweeps of the applied signal [25,26,28].

#### 2.3.2. Electron Microscopy SEM/EDS

At the end of the tests, the steel was extracted from the solution and dried in an oven at 40 °C for a week. Surface morphology was then observed using electron microscopy SEM/EDS. The attacked zone of the galvanized steel was observed by scanning electron microscope. A JEOL 6400 JSM microscope with EDS analysis was used with a resolution of 133 eV.

#### 2.3.3. Optical Microscopy

In addition, an OLYMPUS SZX7 optical microscope with an OLYMPUS SC50 camera was used to characterize and observe the surface of wires after exposure to the corresponding solutions.

## 3. Results and Discussion

### 3.1. LPR Results

Figure 2 shows polarization resistance curves resulting from LPR measurements at the end of the test. Curves show the cathodic and anodic branches and corrosion potential. All zinc wires showed their potentials with an intermediate corrosion probability (E_corr_ > −332 mV) except for Solution 4, which produced somewhat higher values.

Table 2 shows values obtained for E_corr_, R_p_ and I_corr_ of all synthetic solutions made during the entire test period. Wires evolve from high corrosion risk to less electronegative potentials. Figure 3 shows evolution in time of I_corr_ and E_corr_ of the wires.

Starting point results were very electronegative for all solutions (–1.4 V) and showed corrosion current densities around 100 µA/cm^2^. After 35 days of testing, the potential increased to less electronegative values, higher than −0.3 V (except for Solution 4). Corrosion current density decreased to 0.5 µA/cm^2^. These values of E_corr_ were also recorded in other works. In particular, at a pH value of 13, Zn is actively dissolved, giving rise to a soluble phase of Zn(OH)_4_^2−^ in the potential range between −1.35 to −1.45 VSCE [29]. The corrosion current density decreases until they reach values between 0.1 and 1 µA/cm^2^.

The solution containing only sulfate ions SO_4_^2−^ (Solution 1) begins with active I_corr_ (97.24 µA/cm^2^) and a fairly electronegative E_corr_ (−1.4 V). Initial values recorded for E_corr_ and I_corr_ indicate that the galvanized layer upon contact with the alkaline medium dissolves anodically, with consequent evolution of hydrogen on the galvanized surface [6,7]. After ten days of testing, these electrochemical parameters changed. The wire became covered with a passive layer and the hydrogen evolution process slowed down. I_corr_ values decreased and reached 0.28 µA/cm^2^ by the end of the test. This value is within the limits representing a low corrosion state $\left(0.1>\mathrm{Icorr}<0.5\;\mu A/{\mathrm{cm}}^{2}\right)$, as has also been shown in other works [10,12,25] that a passive layer is formed over time. This layer is capable of reducing the initial corrosion current density. This passive layer could be Zn_4_(SO_4_)(OH)_6_ · 3H_2_O [1] in the absence of Ca^2+^ ions.

According to Acha [10] and Liu [11], 0.04 M sulfate ions in saturated Ca(OH)_2_ with a pH value of 12.4 should behave as depassivating ions. An increase of 0.3 units of pH in the presence of sulfates is not enough to passivate the steel. Vigneshwaran et al. [30] considered fixed amounts of sulfate ions of 2000 and 20,000 ppm (0.02 and 0.2 M respectively) to study carbon steel corrosion at pH values of 12.6 and 13.3, respectively. While steel corrodes at a pH of 12.6, at a pH of 13.3 the passive layer is not destabilized. Variation of just one-tenth in the pH value (12.7 vs. 12.6) changes the sulfate ion aggressiveness. With a sulfate ion, the steel is passivated. The corrosion ability of the sulfate ion depends on the pH and the type of steel.

When Ca^2+^ ions (Solution 2) were added to sulfate solution, the initial I_corr_ value of 47.26 µA/cm^2^ was half of the I_corr_ value from Solution 1 and was the lowest value of all solutions. This initial corrosion decreased over time. However, it did so more slowly than in the rest of the solutions. The wire did not reach even a moderate corrosion until the 30th day of testing. The Ca^2+^ concentration was 100 times lower than that of the sulfate solution. The initial concentration of dissolved sulfate decreased by half when precipitating as calcium sulfate. The passive layer on the surface of the steel in the presence of calcium could be different from the previous solution and represent a slower development. Some authors [7,8] identified a passive layer of calcium hydroxyzincate Ca(Zn(OH)_3_)_2_. They indicated a stability limit for this layer at a pH value of 13.3. Above this pH, the larger size of the crystals does not allow them to cover the entire surface of the steel. The I_corr_ value at the end of the test (0.53 µA/cm^2^) was within the limits that represent moderate corrosion $\left(0.5>\mathrm{Icorr}<1\;\mu A/{\mathrm{cm}}^{2}\right)$. At the studied pH, the presence of Ca^2+^ coating turned out to be less protective.

The protection of the wire in the presence of Ca^2+^ ions was modified when NH_4_^+^ ammonium ions were added to the solution (Solution 3). In the beginning, the I_corr_ (92.45 µA/cm^2^) was at least twice that of Solution 2 (47.26 µA/cm^2^) and was very similar to the sulfate solution. H Pan et al. [31] reported than the corrosion of NH_4_^+^ could be attributed to dissolution of the MgO inner layer and the Mg(OH)_2_ outer passive layer. A similar process could take place in the case of ZnO and of Zn(OH)_2,_ increasing the initial corrosion rate. A higher ionic charge increased the solubility of calcium sulfate. The wire began to be covered by a passive layer after fifteen days of testing. The I_corr_ obtained after 35 days reached 0.62 µA/cm^2^, within the limits that represent a state of moderate corrosion $\left(0.5>\mathrm{Icorr}<1\;\mu A/{\mathrm{cm}}^{2}\right)$.

When bicarbonate was added (Solution 4), an initial I_corr_ value of 72.48 µA/cm^2^ was recorded. This was lower than the previous solution but higher than Solution 2. The solubility of zinc carbonate (1.4 · 10^−11^) is lower than that of calcium sulphate (≈9.1 · 10^−6^). Therefore, Zn ions could react with bicarbonate ions according to Equation (5) to produce corresponding carbonates with CO_2_ released [13], thereby removing sulfate ions from the solution.
(5)$${\mathrm{Zn}}^{2+}+2{\mathrm{HCO}}_{3}^{-}↔{\mathrm{ZnCO}}_{3}+H_{2}O+{\mathrm{CO}}_{2}$$

At the end of the test, the I_corr_ (0.73 µA/cm^2^) was the highest of the six solutions and it was within the limits that represent moderate corrosion with low tendency values $\left(0.5>\mathrm{Icorr}<1\;\mu A/{\mathrm{cm}}^{2}\right)$. The corrosion potential shifted to more negative values (−537 mV), which were the most electronegative of the six solutions. A passive film could be formed by hexagonal crystals of ZnCO_3_ and monoclinic crystals of hydrozincite Zn_5_(CO_3_)_2_(OH)_6_. This passive layer would be the least protective.

Subsequently, initial corrosion current density adding Mg^2+^ (Solution 5) was 102 µA/cm^2^. This was higher than previous solutions. After 15 days, the passive layer began to form. I_corr_ values after 35 days (0.19 µA/cm^2^) were within the limits of low corrosion state with a negligible tendency $\left(\;\mathrm{Icorr}<0.1\;\mu A/{\mathrm{cm}}^{2}\right)$. This corrosion current density was the lowest of all solutions, which means that any passive layer formed under these conditions would be the most protective. These results agree with those found by Neupane et al. [21]. They compared the dissolution effects of Na_2_SO_4_, NH_4_SO_4_ and MgSO_4_ on galvanized steel. They found that the corrosion rate in the presence of NH_4_^+^ ions is higher than with Mg^2+^ ions. However, if the pH of the solution is not buffered, the addition of magnesium and ammonium sulfate, accompanied by a decrease in the initial pH of the solution, causes an increase in the corrosion rate, as shown in the study by Xu [20].

Lastly, once the behavior of the wire in chloride-free media was known, its behavior in a medium contaminated with Cl^−^ (Solution 6) was studied. The initial value of I_corr_ was 119 µA/cm^2^, the highest of the six solutions, despite the fact that the concentration of this ion was lower than the sulfate ion solution. I_corr_ values at 35 days (0.42 µA/cm^2^) were higher than the corrosion of the sulfate solution. This was within the limits of low corrosion state with a negligible tendency. A passive layer of simonkolleite Zn_5_Cl_2_(OH)_8_ · H_2_O would thus be less protective than zinc hydroxysulfate (Zn_4_(SO_4_)(OH)_6_ · 3H_2_O.

### 3.2. EIS Results

As noted above, impedance measurement is useful to complete the R_p_ calculation in the linear polarization resistance (LPR) method. In addition, it enables the determination of resistance of the different parts that make up a system.

One of the key aspects of this technique as a tool to research the electrical and electrochemical properties of systems is the direct relationship between the real behavior of a system and that of a circuit made up of a set discrete component of electrical components, called an equivalent circuit. The most accurate circuit will be the one with the fewest possible time constants, which would provide a clear physical meaning [32]. Nyquist and Bode diagrams are obtained with their respective adjustments through the equivalent circuit to determine the values of each parameter that make up the system.

Figure 4 shows the Nyquist and Bode diagram obtained with the adjustment through the equivalent circuit for Solution 1. The equivalent circuit used in this study consisted of two constants connected in series with the resistance of the electrolyte (Figure 5). Elements of the circuit had the following physical meanings: R_s_ was related to the resistance of the electrolyte (solution). Upon the addition of bicarbonate ions (Solution 4), the R_s_ increased due to the greater presence of solid carbonate species in the solution; the first time constants, R_c_ and C_c_, were attributed to the resistance of the passive film on the steel surface [33] and their capacitance. The second time constants, R_ct_ and C_ct_, were related to the charge transfer resistance or mass transfer resistance. The latter was comparable to that obtained by the linear polarization resistance method (R_p_). A constant phase element (*Q*) was used instead of a pure capacitance in the adjustment because of the heterogeneity of the layer on the surface of the wire [32,34]. From the Nyquist diagram, values can be seen corresponding to the high-frequency zone ($1\cdot 10^{5}\;\mathrm{Hz}$), given by the diameter of the first semicircle. It corresponds to R_c_ and C_c_. R_ct_ and C_ct_ correspond to the diameter of the second semicircle in the low-frequency zone (0.01 Hz).

Figure 6 and Figure 7 show the Nyquist diagrams with equivalent circuit adjustments resulting from study cases after 35 days. Parameters obtained from adjustments may be observed in Table 3. From the Nyquist diagrams, it is possible to confirm the beginning of the formation of a passive layer on the surface of the wire. It is also possible to observe the diameter of the second semicircle increasing.

The equivalent circuit used seems to be the correct one because it has the fewest possible time constants with clear physical meaning. The deviation (*χ*^2^) in all cases is less than 0.03. Charge-transfer resistance on the wire surface (R_ct_) is then compared (Table 4) with the obtained R_p_ through linear polarization resistance to validate the results obtained through EIS. Equivalence is maintained in all cases except for Solution 4. The percentage difference between both methods is less than 14%. Although the difference with Solution 4 is 24%, both magnitudes are equal, which represents a state of medium corrosion in both cases $\left(0.5>I_{\mathrm{corr}}<1\;\mu A/{\mathrm{cm}}^{2}\right)$. Differences between outcomes are valid and are mainly attributed to the fact that LPR uses direct current while the EIS technique uses alternating current (sinusoidal disturbance of electric potential) of variable frequency to the studied material). The order of passive layer stability is confirmed by the two electrochemical techniques (Solution 5 < Solution 1 < Solution 6 < Solution 2 > Solution 3 > Solution 4).

### 3.3. Morphology of Steel Surface

Figure 8 shows images of different steel surfaces obtained by optical microscopy and SEM after electrochemical analysis in synthetic solutions. No traces of iron oxides are observed on the surface of the wires. The Solution 1 wire is homogeneously coated with prismatic crystals, possibly of Zn_4_(SO_4_)(OH)_6_ · 3H_2_O. The Solution 3 wire shows a distributed oxide layer, leaving large voids on the surface. The size and coating of the crystals in the rest of the wires varies. This can be attributed to insoluble crystalline products such as Zn_5_(CO_3_)_2_(OH)_6_, ZnCO_3_, Zn_5_Cl_2_(OH)_8_ · H_2_O, ZnO, or others with the ability to passivate galvanized steel to a greater or lesser extent [15]. Some areas of exposure are darker in color due to increased detachment of the zinc layer and oxidation of the steel.

Only one isolated white crystal of calcium hydroxyzincate Ca(Zn(OH)_3_)_2_ was found on the surface of the wire exposed to Solution 2. Due to the high pH of the solution, the crystal becomes larger and does not have the capacity to homogeneously coat the surface of the wire (Figure 9) [14]. By observing the SEM appearance of this crystal, it was possible to identify a totally different morphology from those observed on the surface of the other wires. Small crystallized threads can be observed.

## 4. Conclusions

Galvanized steel in contact with a strongly alkaline solution (pH = 12.7) and in the presence of sulfates dissolves anodically with corrosion potentials around −1.4 V and corrosion densities around 100 µA/cm^2^. Over time, the surface of the steel is covered with a protective layer, and the corrosion potential increases until values of −0.25 V are attained, with a corresponding decrease in corrosion current density to 0.3 µA/cm^2^, but greater than 0.1 µA/cm^2^ in all cases. Consequently, the presence of sulfate ions enables the depassivating of galvanized steel at highly alkaline levels of pH.

The presence of other anions and cations together with the sulfate ions keeps the corrosion process active. The nature of the passive layer depends on the ions present. Cations and anions studied here contribute to the increase in the corrosion current density of the sulfate ions. The magnitude of this increase follows the following orders: NH_4_^+^ > Ca^2+^ > Mg^2+^ and HCO_3_^−^ > Cl^−^ > SO_4_^2−^.

At a pH of 12.7, the hydroxyzincate crystal formed on the surface of the steel immersed in the solution of sulfate and calcium ions is large and occurs in isolation without the ability to cover the entire surface of the steel.

## Figures and Tables

**Figure 1 materials-15-03950-f001:**
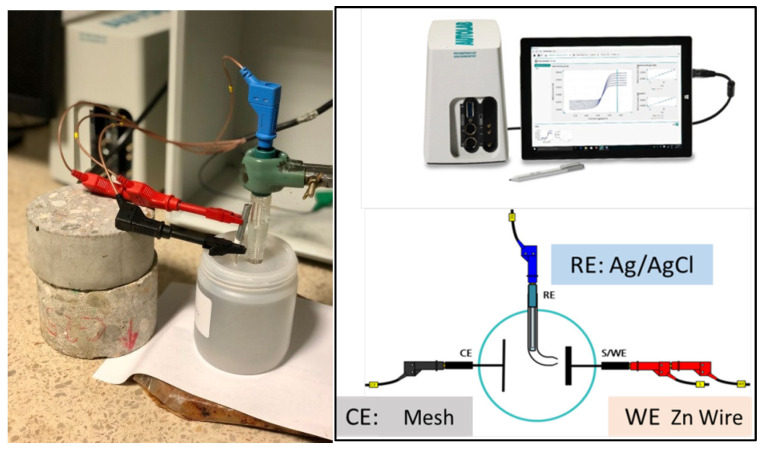
Corrosion cell and Autolab PGSTAT 204 potentiostat/galvanostat assembly together with experimental connections.

**Figure 2 materials-15-03950-f002:**
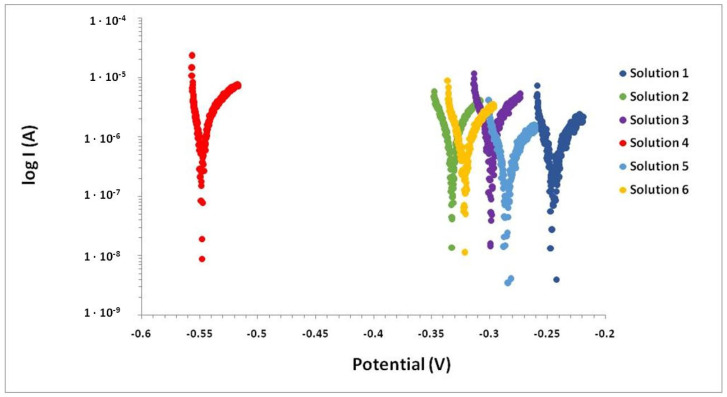
Polarization resistance curves of zinc wires in solution cells after 35 days of manufacture.

**Figure 3 materials-15-03950-f003:**
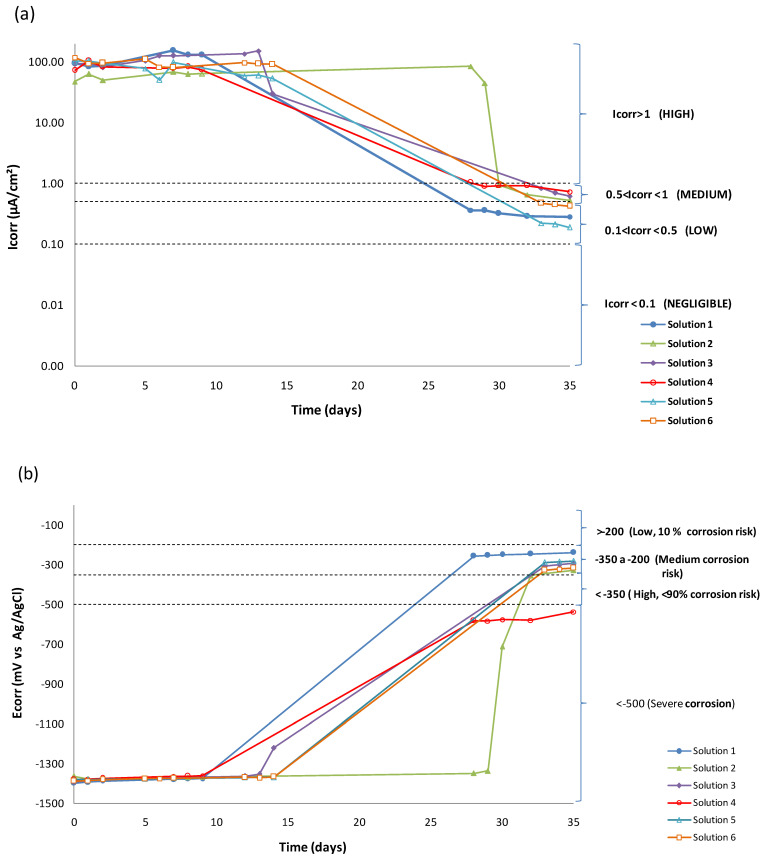
Evolution in time of (**a**) I_corr_ and (**b**) E_corr_ of the wires.

**Figure 4 materials-15-03950-f004:**
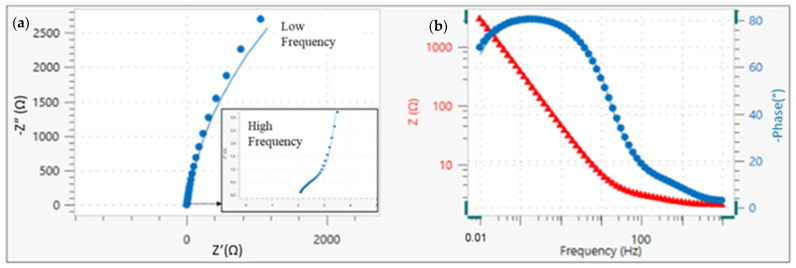
(**a**) EIS results with adjustment through equivalent circuit. (**b**) Nyquist and Bode diagram for Solution 1 at day 35.

**Figure 5 materials-15-03950-f005:**
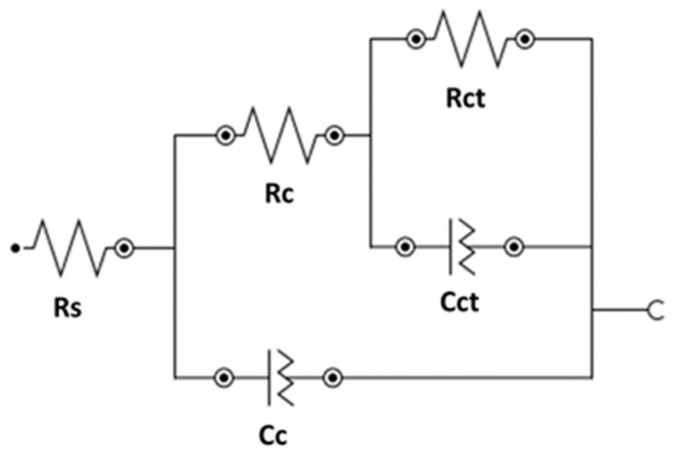
Equivalent circuit.

**Figure 6 materials-15-03950-f006:**
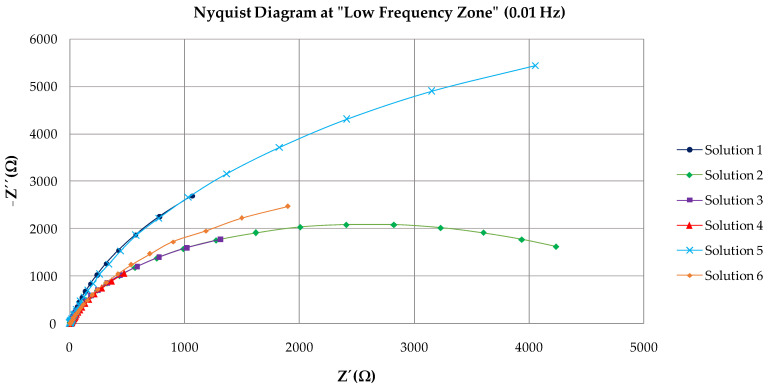
Nyquist diagram in “Low-Frequency Zone” (0.01 Hz).

**Figure 7 materials-15-03950-f007:**
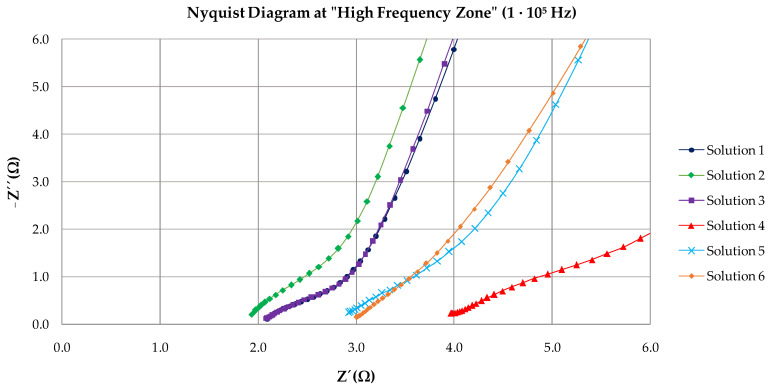
Nyquist diagram in “High-Frequency Zone” (1 · 10^5^ Hz).

**Figure 8 materials-15-03950-f008:**
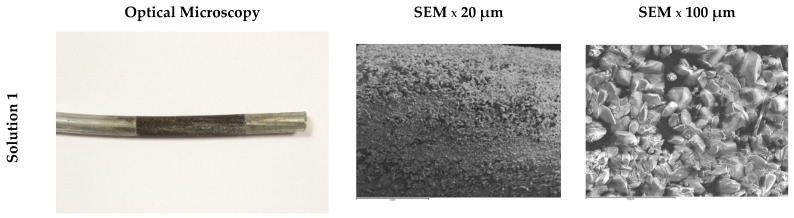
Optical microscopy and SEM results of the steel surfaces.

**Figure 9 materials-15-03950-f009:**
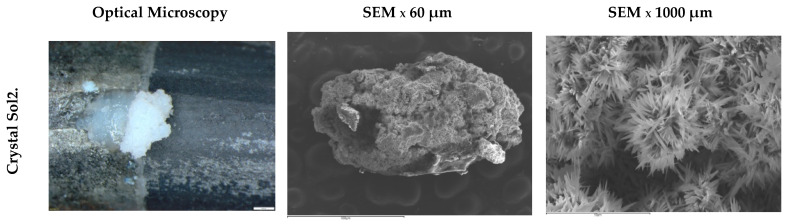
Optical microscopy and SEM crystal results Ca(Zn(OH)_3_)_2_ Solution 2.

**Table 1 materials-15-03950-t001:** Composition of synthetic solutions prepared for corrosion tests.

Solution	SO_4_^2−^ [mol/L] 0.04	Ca^2+^ [mol/L] 4 · 10^−4^	NH_4_^+^ [mol/L] 5 · 10^−3^	HCO_3_^−^ [mol/L] 4 · 10^−4^	Mg^2+^ [mol/L] 9.6 · 10^−3^	Cl^−^ [mol/L] 7.6 · 10^−3^
1	X					
2	X	X				
3	X	X	X			
4	X	X	X	X		
5	X	X	X	X	X	
6	X	X	X	X	X	X

**Table 2 materials-15-03950-t002:** Electrochemical parameters obtained after 35 days of manufacture.

**Day**	**Solution 1**			**Solution 2**			**Solution 3**		
	**E_corr_ (mV)**	**R_p_ (Ω)**	**I_corr_ (µA/cm^2^)**	**E_corr_ (mV)**	**R_p_ (Ω)**	**I_corr_ (µA/cm^2^)**	**E_corr_ (mV)**	**R_p_ (Ω)**	**I_corr_ (µA/cm^2^)**
0	−1403	27.33	97.24	−1373	56.23	47.26	−1396	28.75	92.45
1	−1396	31.45	84.51	−1386	42.26	62.89	−1389	28.30	93.92
8	−1379	20.57	129.20	−1375	55.35	63.35	−1375	21.17	125.54
28	−261	7290.27	0.36	−1351	31.50	84.37	-	-	-
35	−246	9383.37	0.28	−332	5039.88	0.53	−300	4306.20	0.62
**Day**	**Solution 4**			**Solution 5**			**Solution 6**		
	**E_corr_ (mV)**	**R_p_ (Ω)**	**I_corr_ (µA/cm^2^)**	**E_corr_ (mV)**	**R_p_ (Ω)**	**I_corr_ (µA/cm^2^)**	**E_corr_ (mV)**	**R_p_ (Ω)**	**I_corr_ (µA/cm^2^)**
0	−1387	36.67	72.48	−1384	26.07	101.94	−1393	22.43	118.51
1	−1382	25.64	103.66	−1385	25.42	104.54	−1389	28.15	94.40
8	−1367	32.10	82.79	−1379	26.97	98.54	−1378	32.95	80.66
28	−593	2595.41	1.02	-	-	-	-	-	-
35	−548	3660.90	0.73	−287	14,025.21	0.19	−322	6303.55	0.42

**Table 3 materials-15-03950-t003:** Parameters obtained through Equivalent Circuit of the Nyquist Diagram.

Solution	Rs	Rc	CPE	Rct	CPE	χ^2^
	(Ω·cm^2^)	(Ω·cm^2^)	Y_0_(Ω^−1^·cm^−2^·s^n^)	(kΩ·cm^2^)	Y_0_(Ω^−1^·cm^−2^·s^n^)	
1	2.11	1.55	1.92 · 10^−3^	10,169	6.36 · 10^−4^	0.036
2	1.84	2.18	9.20 · 10^−5^	5346	8.59 · 10^−5^	0.024
3	2.13	2.95	4.99 · 10^−4^	4777	3.99 · 10^−4^	0.034
4	3.87	4.48	3.86 ·10^−4^	4665	1.35 · 10^−3^	0.036
5	2.93	2.87	9.20 · 10^−5^	14,923	1.92 · 10^−4^	0.019
6	3.03	4.83	3.21 · 10^−4^	7240	2.51 · 10^−4^	0.016

**Table 4 materials-15-03950-t004:** R_p_ obtained by LPR and EIS.

Solution	R_p_ (LPR)	R_ct_ (EIS)	Difference (%)
	(Ω)	(Ω)	
1	9383	10,169	8
2	5040	5346	6
3	4306	4777	10
4	3661	4665	24
5	14,025	14,923	6
6	6304	7240	14

## Data Availability

Not applicable.

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
