# Peer review of "Effect of Sulfate Ions on Galvanized Post-Tensioned Steel Corrosion in Alkaline Solutions and the Interaction with Other Ions"

_materials, 2022, doi:10.3390/ma15113950_

Round 1

Reviewer 1 Report

This manuscript investigates the effect of some anions on the corrosion behavior of galvanized steel wire. Huge data are presented, and different anions are analyzed. However, there exist some problems in the manuscript. It needs a through revision before it can be accepted in this journal.

  1. Please draw the figures in a commercial software rather than only using the screenshot images. For the polarization curves and EIS, they like to be obtained from the autolab software.
  2. Using corrosion current density instead of corrosion density.
  3. How do you conduct the long-term polarization curves? For this technique, it belongs to a destructive method and the specimens cannot be used for the continuous immersion. So if you conduct polarization curves on the samples after immersion for different days, the different samples should be used.
  4. The equivalent circuit. What is the Rc? If it is charge transfer resistance, then what is the Rct?
  5. Moreover, in you case, the Rc value is so small. What is the physical meaning of such small resistance? Maybe it is more proper to use the circuit with only one time constant.
  6. The unit in Table 3 are all wrong. Please refer to Corrosion Science 150 (2019) 218–234 to revise it.
  7. The results show that the increase magnitude is the highest for the NH4+ anion. It is better for the authors to give a brief analysis of this effect. The roles of NH4+ in this system can be referred to Journal of Materials Science & Technology 54 (2020) 1–13 and Electrochimica Acta 278 (2018) 421-437.

Author Response

Dear Reviewer,

We would like to sincerely thank the reviewers for their comments, which have helped us to improve the quality of the paper, we have found their comments helpful and constructive.

Below we address each of the remarks made by the authors (in blue) to the comments of the reviewers and reference the changes made in the manuscript (highlighted in yellow in the new version).

This manuscript investigates the effect of some anions on the corrosion behavior of galvanized steel wire. Huge data are presented, and different anions are analyzed. However, there exist some problems in the manuscript. It needs a through revision before it can be accepted in this journal.

  • Please draw the figures in a commercial software rather than only using the screenshot images. For the polarization curves and EIS, they like to be obtained from the autolab software.

The authors agree the comment of the reviewer. EIS graphics supplied by Autolab software were changed using the original experimental data recorded in Excel spreadsheets.

  • Using corrosion current density instead of corrosion density.

Text has been corrected.

  • How do you conduct the long-term polarization curves? For this technique, it belongs to a destructive method and the specimens cannot be used for the continuous immersion. So if you conduct polarization curves on the samples after immersion for different days, the different samples should be used.

The correct term of the non destructive technique is “polarization resistance curve “, which has been included in the manuscript replacing “polarization curve”, following the reviewer request.

  • The equivalent circuit. What is the Rc? If it is charge transfer resistance, then what is the Rct? Moreover, in you case, the Rc value is so small. What is the physical meaning of such small resistance? Maybe it is more proper to use the circuit with only one time constant.

Thank you for your suggestion.

The second resistance Rc is attributed to the resistance of the passive film on the steel surface. This point has been changed in the manuscript. The value of Rc is low due to the process of solution and in this medium is reduced.

The authors agree with the reviewer. It could be used the circuit with only one time constant, by adding two resistances. It was performed by the authors, but the fitting was worse, reaching larger χ2 error. The best fit was obtained by separating both resistances.

  • The unit in Table 3 are all wrong. Please refer to Corrosion Science 150 (2019) 218–234 to revise it.

Units of Table 3 have been corrected according to the CPE value.

  • The results show that the increase magnitude is the highest for the NH4+ anion. It is better for the authors to give a brief analysis of this effect. The roles of NH4+ in this system can be referred to Journal of Materials Science & Technology 54 (2020) 1–13 and Electrochimica Acta 278 (2018) 421-437.

A brief analysis of NH4+ anion is included in the manuscript

Yours faithfully,

The authors

Reviewer 2 Report

The authors studied the corrosion properties of Zn-coated steel in different solutions. Some question are listed as below.

  1. The conditions of experimental alloy should be included in the manuscript, such as the compositions of the steel, the thickness of the Zn layer, and how to get (or prepare) the experimental alloy.
  2. Why did the authors test in strongly alkaline solutions (pH=12.7)? Also, what field (or environment) can the alloy be applied?
  3. What is the unit of Ecorr (V vs SHE or SCE)?
  4. The micron bars in Figure 8 and Figure 9 are too small.

Author Response

Dear Reviewer,

We would like to sincerely thank the reviewers for their comments, which have helped us to improve the quality of the paper, we have found their comments helpful and constructive.

Below we address each of the remarks made by the authors (in blue) to the comments of the reviewers and reference the changes made in the manuscript (highlighted in yellow in the new version).

The authors studied the corrosion properties of Zn-coated steel in different solutions. Somequestion are listed as below.

  • The conditions of experimental alloy should be included in the manuscript, such as the compositions of the steel, the thickness of the Zn layer, and how to get (or prepare) the experimental alloy.

The authors agree with the reviewer, but the material was commercially supplied without this information and, unfortunately, it was not possible to obtain the galvanized steel chemical composition.

  • Why did the authors test in strongly alkaline solutions (pH=12.7)? Also, what field (or environment) can the alloy be applied?

The tests reproduce the real conditions (on site) to which the galvanized steels were exposed to study the steel corrosion. The synthetic solutions also reproduce the ions and pH found in the actual solutions, which is some cases are in contact with the galvanized steels.

  • What is the unit of Ecorr (V vs SHE or SCE)?

Ecorr units are mV (mV vs AgCl/Ag), as is indicated in figure 3b).

  • The micron bars in Figure 8 and Figure 9 are too small.

Figures have been corrected.

Yours faithfully,

The authors

Reviewer 3 Report

Below you can find the list of the manuscript’s weaknesses.

  1. The Introduction is lacking univocally written objectives of the study with stressed out scientific importance.
  2. in the equation (1) are present some extra dash signs which should not be there.
  3. The equation 5 has wrong stoichiometry – that needs to be corrected

I suggest accepting this manuscript for publication after correction of above.

Author Response

Dear Reviewer,

We would like to sincerely thank the reviewers for their comments, which have helped us to improve the quality of the paper, we have found their comments helpful and constructive.

Below we address each of the remarks made by the authors (in blue) to the comments of the reviewers and reference the changes made in the manuscript (highlighted in yellow in the new version).

Below you can find the list of the manuscript’s weaknesses.

  • The Introduction is lacking univocally written objectives of the study with stressed out scientific importance.

Thank you for your suggestion. The authors have clarified the objectives as follows:

In the present work is studied the effect of the sulfate ion in alkaline solution in galvanized steel and the synergistic effect of sulfate ion with various cations and anions found in seawater or marine environments (Ca2+, NH4+, HCO3-, Mg2+ and Cl-) or in the atmosphere and the influence on the corrosion of the galvanized steel wires.

  • in the equation (1) are present some extra dash signs which should not be there.

The extra dash signs in equation (1) have been removed.

  • The equation 5 has wrong stoichiometry – that needs to be corrected.

The stoichiometric adjustment of equation 5 has been made.

I suggest accepting this manuscript for publication after correction of above.

Thank you

Yours faithfully,

The authors

Round 2

Reviewer 1 Report

It can be accepted in the present form.

Reviewer 2 Report

I agree with this version of manuscript.